

# Quantum scars and caustics in Majorana billiards

R. Johanna Zijderveld[1*], A. Mert Bozkurt[1,2†],
Michael Wimmer[1,2] and İnanç Adagideli[3,4,5‡]

**1** Kavli Institute of Nanoscience, Delft University of Technology,
P.O. Box 4056, 2600 GA Delft, The Netherlands
**2** QuTech, Delft University of Technology, P.O. Box 4056, Delft 2600 GA, The Netherlands
**3** Faculty of Engineering and Natural Sciences, Sabanci University, 34956 Istanbul, Turkey
**4** MESA+ Institute for Nanotechnology, University of Twente,
7500 AE Enschede, The Netherlands
**5** TÜBİTAK Research Institute for Fundamental Sciences, 41470 Gebze, Turkey

⋆ johanna@zijderveld.de , † a.mertbozkurt@gmail.com , ‡ adagideli@sabanciuniv.edu

## Abstract

We demonstrate that the classical dynamics influence the localization behaviour of Majorana wavefunctions in Majorana billiards. By using a connection between Majorana wavefunctions and eigenfunctions of a normal state Hamiltonian, we show that Majorana wavefunctions in both p-wave and s-wave topological superconductors inherit the properties of the underlying normal state eigenfunctions. As an example, we demonstrate that Majorana wavefunctions in topological superconductors with chaotic shapes feature quantum scarring. Furthermore, we show a way to manipulate a localized Majorana wavefunction by altering the underlying classical dynamics using a local potential away from the localization region. Finally, in the presence of chiral symmetry breaking, we find that the Majorana wavefunction in convex-shaped Majorana billiards exhibits caustics formation, reminiscent of a normal state system with magnetic field.

# 1  Introduction

Over the past decade, search of Majorana zero modes in topological superconductors has attracted significant interest within the condensed matter community [1–3]. Majorana zero modes are anyons governed by non-Abelian braiding statistics [4]. The spatial separation of Majorana zero modes protects the quantum information encoded by these modes from decoherence while maintaining its robustness due to their non-local nature. This separation allows Majorana zero modes to be used for the creation of topologically protected qubits [5]. Therefore, understanding the localization properties of Majorana zero modes in topological superconductors within different scenarios and geometries holds significant importance.

In a topological superconductor, such as for example a p-wave superconductor [6], the superconducting pairing potential sets the localization properties of Majorana zero modes [7]. As the magnitude of the pairing potential increases, Majorana zero modes exhibit stronger localization towards the edges of the topological superconductor [8]. In contrast to topologically trivial disordered systems, increasing disorder strength in topological superconductors *increases* the localization length of the Majorana zero modes [9,10]. As a consequence, there is a transition from topological phase to Anderson insulator phase at a critical disorder strength [11] and there are no Majorana zero modes present.

While pairing potential and disorder effects are the conventional sources of localization, ballistic systems which feature chaotic dynamics in their classical limit also show localization, which we call ballistic chaotic localization. One consequence of such localization is the ballistic weak localization [12–16] correction in the conductance of clean chaotic cavities. Ballistic chaotic systems also include Andreev billiards [17,18], where such localization phenomena have been theoretically studied. Another consequence of ballistic chaotic localization is that, in the small wavelength limit, the eigenfunctions of a quantum system exhibit quantum scarring [19,20]. In this limit, local perturbations can also induce quantum scarring [21–23]. Quantum scarring can be viewed as the localization of the eigenstates of a system with chaotic dynamics along an unstable periodic orbit [24,25]. Quantum scars have been observed in diverse settings, including microwave cavities [26,27], optical cavities [28,29], and quantum wells [30,31]. Additionally, the notion of "quantum many-body scarring" [32,33] has been put forth as a possible explanation for the experimentally observed phenomenon of slow thermalization in cold atoms [34].

Inspired by these additional mechanisms of chaotic localization in ballistic systems, we pose the following question: "how do the classical dynamics affect the localization of Majorana wavefunctions?". To answer this question, we explore the connection between the classical dynamics of the normal metallic phase and the Majorana wavefunctions in the topological superconducting phase of the same shape. In particular, we focus on two seminal examples from semiclassical physics: quantum scarring and caustics. We find that traces of quantum scars occur in both p-wave and s-wave topological superconductors. Remarkably, we find that it is possible to take advantage of this connection to manipulate the position of the Majorana zero modes by changing the classical dynamics, such as by introducing local potentials or controlling the convexity of the shape.

This manuscript is structured as follows: In Section 2, we demonstrate how the Majorana wavefunction of a chiral symmetric spinless p-wave topological superconductor is mapped to a normal state eigenfunction. Building upon this mapping, Section 3 explores the phenomenon of quantum scarring in the Majorana wavefunction of classically chaotic topological superconductors. In Section 4, we investigate the influence of local potentials located far from the system's edges on the Majorana wavefunction. Moving forward, Section 5 considers chiral symmetry breaking and its implications for caustics formation in the Majorana wavefunction. For completeness, Section 6 extends our analysis to a more realistic system, characterized by an s-wave superconductor with Rashba spin-orbit coupling and Zeeman energy, confirming the generality of our findings. Finally, in Section 7, we provide a concise overview and conclusion.

## 2 Mapping the fermion parity switch to a normal state eigenvalue equation

We first consider the minimal model which describes a topological superconductor: a spinless p-wave Hamiltonian in two dimensions. The Hamiltonian is given by [2]:

$$H_p = \left(\frac{\mathbf{p}^2}{2m} - \mu\right)\tau_z + \Delta_x p_x \tau_x + \Delta_y p_y \tau_y + V(\mathbf{r}), \tag{1}$$

where $\tau$ are the Pauli matrices in the particle-hole basis, $\mu$ is the chemical potential of the p-wave Hamiltonian, $\mathbf{p}$ is the momentum operator, $m$ is the effective mass and $\Delta$ is the p-wave superconducting pairing term. $V(\mathbf{r})$ is the confinement potential, with hard wall boundary conditions, that determines the shape of the topological superconductor, with $\mathbf{r} = (x, y)$. The zero energy solutions of Eq. (1) describe Majorana zero modes in 1D and Majorana fermions in 2D. We look for these zero energy solutions by solving:

$$H_p|_{\mu=\mu_p}\Psi = 0, \tag{2}$$

where $\mu_p$ is the chemical potential value at a fermion parity switch. The solution to Eq. (2) can be mapped onto a non-Hermitian eigenvalue problem with real eigenvalues which are the chemical potential values of fermion parity switches [35]. To show this, we start by premultiplying Eq. (2) with $\tau_z$ and move the $\mu_p$ chemical potential to the right hand side:

$$\left(\frac{(\mathbf{p} + im\boldsymbol{\eta})^2}{2m} + \frac{m}{2}(\Delta_x^2 + \Delta_y^2) + V(\mathbf{r})\right)\tau_0\Psi = \mu_p \tau_0 \Psi, \tag{3}$$

where $\boldsymbol{\eta} = \Delta_x \tau_y \hat{x} - \Delta_y \tau_x \hat{y}$. Now the operator on the left is non-Hermitian as a result of multiplication with $\tau_z$, with the term $im\boldsymbol{\eta}$ resembling a Rashba spin-orbit coupling with an imaginary Rashba parameter.

We now make a distinction between the topological superconductors with or without chiral symmetry. When both $\Delta_x > 0$ and $\Delta_y > 0$, there is no chiral symmetry as there is no operator which anticommutes with $H_p$. However, in the presence of chiral symmetry, the non-Hermitian operator in Eq. (3) becomes a Hermitian operator via an imaginary gauge transformation. Therefore, we first consider this chiral symmetric case, in which we take the convention that $\Delta_y = 0$ and $\Delta_x > 0$. This scenario becomes applicable when the confinement in the $y$-direction is smaller than the coherence length [36]. This then allows us to gauge away the $\boldsymbol{\eta}$ term in Eq. (3).

After performing the gauge transformation, we obtain the following Hermitian eigenvalue equation:

$$\left(\frac{\mathbf{p}^2}{2m} + \frac{1}{2}m\Delta_x^2 + V(\mathbf{r})\right)\tau_0\tilde{\Psi} = \mu_p \tau_0 \tilde{\Psi}, \tag{4}$$

where the rescaled wavefunction $\tilde{\Psi}$ obeys the following equation:

$$\Psi = \mathcal{N} e^{\tau_y x/\xi} \tilde{\Psi}, \tag{5}$$

where we define the superconducting coherence length as $\xi = \frac{\hbar}{m\Delta_x}$. Here, $\mathcal{N}$ is the normalization factor, which is defined as

$$\mathcal{N}^{-2} = \int d\mathbf{r}\, e^{\pm 2x/\xi} |\tilde{\Psi}(\mathbf{r})|^2, \tag{6}$$

where the integral is over the whole billiard and $\pm$ sign in the exponential function depends on the $\tau_y$ eigenvalue.

The Hermitian operator on the left hand side of Eq. (4) is a normal state Hamiltonian with a constant energy shift $m\Delta_x^2/2$, multiplied by identity operator in particle-hole basis. As a result, obtaining either particle or hole component of $\tilde{\Psi}$ is equivalent to solving for the eigenfunctions $\Psi_n$ of the following normal state Hamiltonian:

$$\left(\frac{\mathbf{p}^2}{2m} + V(\mathbf{r})\right)\Psi_n = \epsilon_n \Psi_n, \tag{7}$$

where $\epsilon_n \equiv \mu_p - m\Delta_x^2/2$. Following Eq. (5) and the correspondence between $\tilde{\Psi}$ and $\Psi_n$, we conclude that the Majorana wavefunction $\Psi$ at the fermion parity switch is identical to the underlying normal state eigenfunction $\Psi_n$, up to the rescaling given in Eq. (5). This correspondence also holds when we move slightly away from the fermion parity switch.

The rescaling is responsible for pushing the Majorana wavefunctions towards the edges of the topological superconductor along the $x-$direction with a strength proportional to $\Delta_x$. In contrast, the remaining part of the Majorana wavefunction $\tilde{\Psi}$ is identical to $\Psi_n$ and therefore solely determined by the confinement potential $V(\mathbf{r})$, i.e. the shape of the topological superconductor.

To showcase this correspondence between Majorana and regular wavefunctions, we perform tight-binding calculations on a square grid using the computational package Kwant [37]. The conceptual algorithm we use in order to obtain corresponding wavefunctions is as follows:

1. We first solve for the eigenfunctions of a normal state Hamiltonian in a shape determined by the confinement potential $V(\mathbf{r})$.

2. We choose the normal state eigenfunction we are interested in and its corresponding eigenvalue $\epsilon_n$.

3. For a given pairing term $\Delta_x$, we set the superconducting chemical potential as $\mu_p = \epsilon_n + m\Delta_x^2/2$.

4. Following this, we solve for the zero energy solutions of a p-wave Hamiltonian with the same confinement potential $V(\mathbf{r})$ as the regular Hamiltonian.

5. The resulting eigenstate for the superconducting system $\Psi$ has the same local profile as the normal state eigenfunction up to the rescaling.

In Fig. 1, we show an example of this conceptual algorithm. Fig. 1a) shows a normal state eigenfunction confined in the shape of a half-stadium billiard. Then, for a particular superconducting pairing term $\Delta_x$, we shift the chemical potential $\mu_p$ and solve for the Majorana wavefunction of the topological superconductor with the same shape. The resulting Majorana wavefunction, as shown in Fig. 1b), has the same pattern as the normal wavefunction in Fig. 1a). However, the Majorana wavefunction has less amplitude in the center of the billiard than the normal state wavefunction.

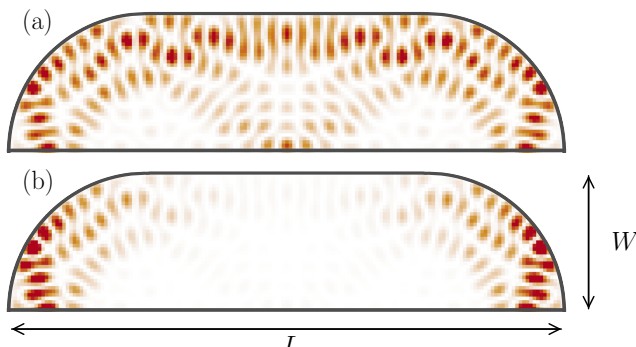

Figure 1: Correspondence between the eigenfunctions of a normal state Hamiltonian and the Majorana wavefunction of a p-wave topological superconductor in a half-stadium billiard with dimensions $L = 160a$, $W = 40a$. (a) The wavefunction density of a normal Hamiltonian with $\mu = 0.58t$ and $t = 1$. (b) The corresponding Majorana wavefunction density, $\rho(\mathbf{r}) = \Psi^\dagger(\mathbf{r})\Psi(\mathbf{r})$, with the superconducting pairing term at $\Delta_x = 0.06t$.

## 3 Quantum scarring in the Majorana wavefunction

Inspired by the correspondence between the eigenfunctions of a regular Hamiltonian and the Majorana wavefunction of p-wave Hamiltonian, we consider both a regular and p-wave Hamiltonian in a chaotic billiard in the semiclassical limit. In this limit, where the system size is much larger than the Fermi wavelength $\lambda_F = h/\sqrt{2m\mu}$, the eigenstates of the regular Hamiltonian in a chaotic billiard exhibit quantum scarring. We observe traces of these quantum scars in the Majorana wavefunction by following the conceptual algorithm we provided in the previous section.

We use the inverse participation ratio (IPR), $\sum_i |\Psi(x_i)|^4$, as a measure of the localization of the normal state wavefunction to choose the scarred eigenstates of the normal Hamiltonian [38, 39]. Specifically, we numerically calculate the IPR for each eigenstate of the normal Hamiltonian, and choose the one with highest IPR. If the local density of states of the normal state eigenfunction is scarred, we then follow the next steps of our conceptual algorithm and obtain the Majorana wavefunction.

We show an example of this procedure in Fig. 2. Fig. 2a) and b) show the local density of states of a scarred regular eigenfunction and corresponding Majorana wavefunction in a stadium billiard. We transformed the Majorana wavefunction to be in the $\tau_y$ basis, which causes it to localize in only one edge of the system. We performed this transformation in order to show the scaling behaviour described in Eq. (5). Fig. 2c) shows the localization of the Majorana wavefunction as a function of the pairing potential $\Delta_x$. To this end, we plot the logarithm of the normalized overlap between the Majorana wavefunction at $\Delta_x = 0$ and the Majorana wavefunction at finite $\Delta_x$ for different $x$ values in the stadium billiard:

$$\log\left(\langle\tilde{\Psi}(x)|\Psi(x)\rangle\right) \equiv \log\left(\frac{\langle\Psi_{\Delta_x=0}(x)|\Psi(x)\rangle}{\mathcal{N}(\Delta_x)\langle\Psi_{\Delta_x=0}(x)|\Psi_{\Delta_x=0}(x)\rangle}\right). \tag{8}$$

As expected from Eq. (5), this overlap perfectly follows the analytical $\log(e^{x/\xi})$ lines. Thus, we conclude that the Majorana wavefunction shows quantum scarring at the edges of a topological superconductor chaotic billiard in the semiclassical regime. As quantum scars depend directly on the classical dynamics, i.e. on the shape of the system, so then does the localization of the Majorana wavefunction.

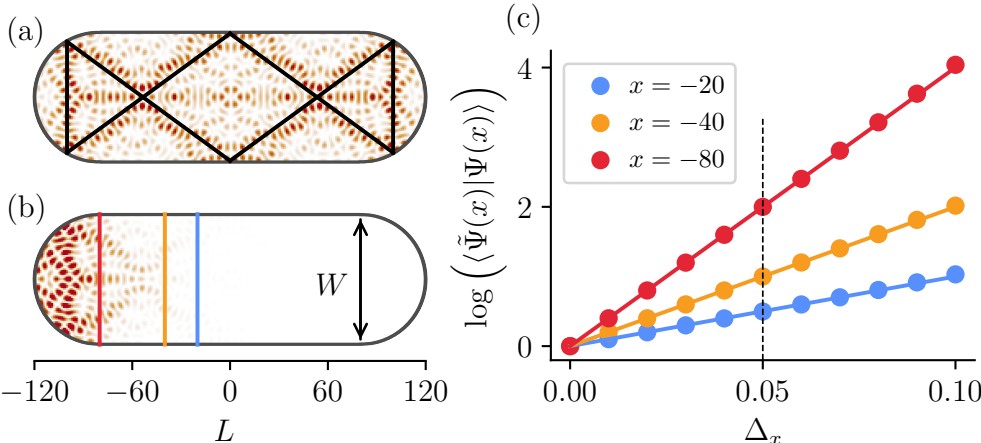

Figure 2: Quantum scarring in regular and Majorana wavefunctions. (a) shows the wavefunction density in billiard governed by a normal Hamiltonian with $\mu = 0.6t$ and the dimensions $L = 240a, W = 80a$. The path of the scar in the normal billiard is shown by the black line. (b) The corresponding Majorana wavefunction density with the superconducting pairing term at $\Delta_x = 0.05t$. (c) The log of the normalized overlap between the Majorana wavefunction at $\Delta_x = 0$ and the Majorana wavefunction at finite $\Delta_x$ of the billiard in (b), taken on the lines at $x = -20a, -40a, -80a$.

We confirm that classical dynamics influence the localization of the Majorana wavefunction by quantitatively analyzing the IPR of scarred and non-scarred eigenstates at increasing $\Delta_x$ values. First, we select scarred and non-scarred eigenstates of the normal Hamiltonian based on high and low IPR values, respectively. Following the theoretical steps above, we obtain the corresponding Majorana wavefunctions $\Psi$ for different $\Delta_x$ values and calculate their IPR. Additionally, we compute the IPR of the Majorana wavefunctions semi-analytically, i.e. by using a rescaled normal state wavefunction, as defined in Eq. (5).[1] We also apply this scaling to an artificially constructed uniform normal state wavefunction. Fig. 3 shows that the localization of the Majorana wavefunction, quantified by the IPR, depends on the underlying normal state wavefunction, as the IPR vs. $\Delta_x$ dependence is drastically different for scarred and non-scarred wavefunctions. This reveals how quantum scarring can help the Majorana wavefunction to localize further. Moreover, we observe that the IPR dependence for both scarred and non-scarred states differs from that of the uniform normal state wavefunction.[2] We thus show that the localization properties of the underlying normal state wavefunctions significantly influence the localization of the corresponding Majorana wavefunctions.

## 4   Manipulating the Majorana localization by a stopper

As the Majorana wavefunction is inherently linked to the normal state wavefunction, we ask the following question: What happens to the Majorana wavefunction when we introduce a local perturbation away from the typical regions of localized Majoranas? This question is intriguing as Majoranas are generally expected to be immune to local perturbations.

---

[1]While deriving our mapping, we assume a continuum Hamiltonian. However, tight-binding simulations suffer from discretization errors at large $\Delta_x$ and deviate from the continuum description. For this reason, we compare semi-analytical results with tight-binding results here.

[2]As $\Delta_x$ localizes the normal state wavefunction near the system edges, a non-scarred Majorana wavefunction initially with low IPR could show higher IPR at specific $\Delta_x$ values compared to a scarred counterpart.

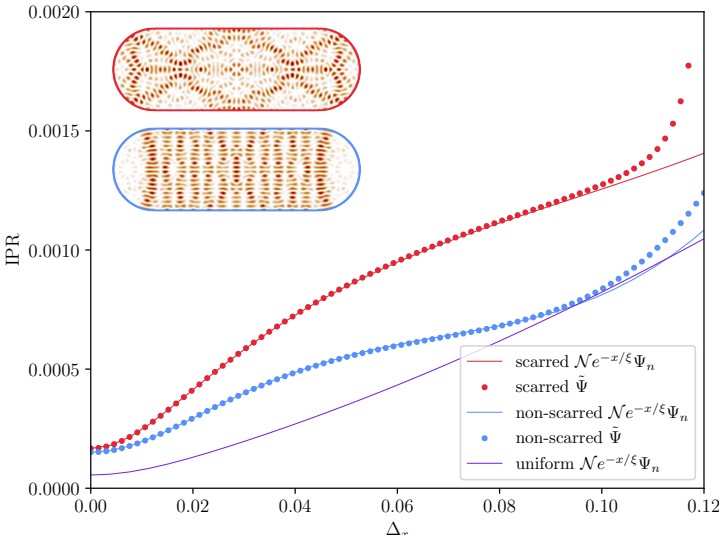

Figure 3: TThe IPR dependence of a scarred (red) and non-scarred (blue) Majorana wavefunction for different $\Delta_x$ values is shown. The lines represent the semi-analytical Majorana wavefunctions, constructed by scaling the numerically obtained normal state wavefunctions using Eq.(5), while the dots represent the Majorana wavefunctions obtained by solving the tight-binding p-wave Hamiltonian in Eq. (1). Furthermore, the analytical IPR dependence of a uniform wavefunction is shown (in purple). The insets show the local density profile of the scarred (red boundary) and non-scarred (blue boundary) normal state wavefunctions. The Majorana wavefunctions were calculated at $\mu = 0.6t$ in a billiard of dimensions $L = 240a, W = 80a$.

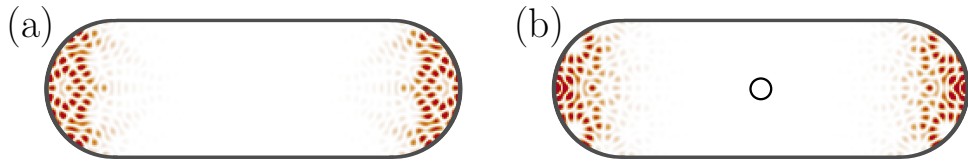

Figure 4: Influence of a stopper on the Majorana wavefunction in a stadium billiard of dimensions $L = 240a, W = 80a$. Panel (a) shows the wavefunction density of a p-wave Hamiltonian with $\mu = 0.6t$ and $\Delta_x = 0.1$ in the absence of a stopper. Panel (b) shows the wavefunction of a Hamiltonian with the same parameters, but with a stopper of strength $2t$ and size $r = 6a$ at $(x, y) = (0, 0)$, as depicted by the black circle.

To answer this question, we consider a case where we introduce a stopper, which is a local increase in onsite potential, in the middle of a stadium-shaped 2d p-wave superconductor. We first show a scarred Majorana wavefunction without a stopper, at large value of of $\Delta_x$ in Fig. 4a). This Majorana wavefunction is highly localized near the edges of the billiard. Hence, a stopper in the middle of the billiard remains far away from the regions of localization, with the expectation that it will not affect the Majorana wavefunction.

However, as we show in Fig. 4b), the local profile of the Majorana wavefunction does change due to the presence of a stopper. The Majorana wavefunction is still localized near the edges, but its local profile is not the same as the Majorana wavefunction without the stopper shown in Fig. 4a).

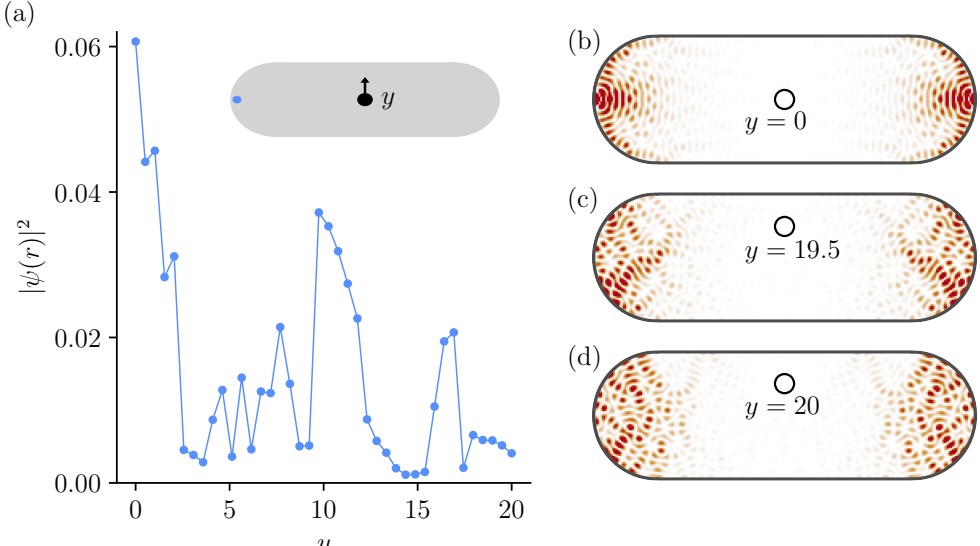

Figure 5: The change in the Majorana wavefunction due the movement of a stopper in the middle of a stadium billiard with dimensions $L = 240a, W = 80a$ and as parameters $\mu = 0.60t$ and $\Delta_x = 0.06ta$. (a) The density squared inside the blue circled region, shown in the grey inset, as a function of $y$ position of a stopper, shown by the black circle in the inset. (b), (c), (d) The full wavefunction density when the stopper is at the positions $y = 0, 19.5a, 20a$ respectively.

We understand this change in the Majorana wavefunction through the previously discussed relation between regular wavefunctions and Majorana wavefunctions. A stopper in the middle of a normal state billiard would alter the possible trajectories of the billiard, which then results in a different normal state eigenfunction. As the local density of the Majorana wavefunction matches the normal state eigenfunctions of the same billiard, the change in normal state eigenfunction due to a stopper indirectly explains the change in the Majorana wavefunction.

The Majorana wavefunction exhibits varying local features in response to changes in the local potential. We investigate this in Fig. 5 by placing the stopper in different positions at the center of the billiard and observing the resulting differences in the Majorana wavefunction. Fig. 5a) shows that stoppers with slightly differing positions cause large changes in the wavefunction amplitude at the very edge of the billiard.

Given this sensitivity of the local features of Majorana wavefunctions due to the position of the stopper potential, we propose an experimental setup to empirically verify this behavior. To determine whether the Majorana wavefunction has a nonzero amplitude at specific points, we propose to employ transport measurements at both ends of the topological superconductor while varying the position of the STM tip. As the STM tip functions as a local potential, varying its position can lead to changes in the differential conductance measurements taken at the edges of the topological superconductor.

## 5 Caustic formation in the Majorana wavefunction

We now consider the p-wave topological superconductor Hamiltonian with broken chiral symmetry, as shown in Eq. (1). In this case, both components of the superconducting pairing term, $\Delta_x$ and $\Delta_y$ are non-zero. As a result, the gauge transformation in our previous derivation of Eq. (4) needs to be extended in order to incorporate the chiral symmetry breaking term.

We can achieve this by treating the chiral symmetry breaking term, either $\Delta_x$ or $\Delta_y$ as a perturbation. This implies that the size of the system along that particular direction has to be smaller than the superconducting coherence length given by that corresponding pairing term, $L_i < \hbar/(m\Delta_i) \equiv \xi_i$. In this limit of weak chiral symmetry breaking, we apply another imaginary gauge transformation to Eq. (3) and arrive at the following eigenvalue equation (see Appendix A for details)

$$\left( \frac{(\mathbf{p} + e\mathbf{A}(\mathbf{r})\tau_z)^2}{2m} + \frac{m}{2}\left( \Delta_x^2 + \Delta_y^2 \right) \right) \tilde{\Psi} = \mu_p \tilde{\Psi}, \qquad (9)$$

where we define $\mathbf{A}(\mathbf{r}) \equiv \frac{2m^2 \Delta_x \Delta_y}{e\hbar}(\hat{\mathbf{z}} \times \mathbf{r})$. This $\mathbf{A}(\mathbf{r})$ term is analogous to a vector potential, equivalent to an out-of-plane magnetic field in the corresponding normal Hamiltonian eigenvalue equation. This fictitious magnetic field acts on electrons and holes with opposite signs, as indicated by the Pauli matrix $\tau_z$. The eigenfunctions of the eigenvalue equation given in Eq. (9) is related to the Majorana wavefunction by

$$\Psi = e^{x/\xi_x \tau_y - y/\xi_y \tau_x} e^{-x^2/\xi_x^2 - y^2/\xi_y^2} \tilde{\Psi}. \qquad (10)$$

Similar to the case with chiral symmetry, the Majorana wavefunction is related to a normal state eigenfunction through Eq. (10). However, the key difference is that the normal state Hamiltonian now features a magnetic field-like term specified by the vector potential $\mathbf{A}(\mathbf{r})$.

We now study the consequences of breaking chiral symmetry in a 2D p-wave topological superconductor, as such a magnetic field-like term might influence the localization, depending on the shape of a billiard. We first consider a convex shaped billiard, comprised of two circles with an offset as shown in Fig. 6, and display the Majorana wavefunction for various $\Delta_y$ values. We observe that as soon as the chiral symmetry is broken, shown in Fig. 6(b-c), the Majorana wavefunction exhibits focusing effects. In contrast, for chiral symmetric case as shown in Fig. 6(a), this feature is absent. These increased local density regions shown in Fig. 6(b,c) are reminiscent of caustics found in semiclassical physics. Berry has shown that the presence of a perpendicular uniform magnetic field results in the formation of such caustics in convex-shaped billiards [40]. Moreover, the form of our effective Hamiltonian given in Eq. (9) suggests a connection between the appearance of caustics and the fictitious magnetic field. However, the effective cyclotron radius resulting from this magnetic field is much larger than the system size, thus it cannot directly account for the observed caustic features in the Majorana wavefunction (see Appendix A for more details). Thus, we conjecture that the appearence of caustics is primarily a consequence of chiral symmetry breaking.

To investigate whether the chiral symmetry breaking has any influence on the relation between the shape of a topological superconductor and the Majorana localization, we consider two distinct types of geometries: convex and non-convex shapes. Fig. 7(a-b) reveals that a convex-shaped topological superconductor exhibits increased local density precisely at locations where classical trajectories intersect due to its convex shape. Conversely, in the case of non-convex topological superconductors, there is no such phenomenon observed in the Majorana wavefunction, as depicted in Fig.7(d-e). In these non-convex topological superconductors, the Majorana wavefunction predominantly concentrates its local density in the corners.

As we increase the pairing $\Delta$ further, the focusing effect vanishes for the convex-shaped topological superconductors. Moreover, both convex and non-convex superconductors reveal the presence of the chiral edge state wavefunction in this limit [41], as depicted in Fig. 7(c,f). We emphasize that this is expected, as the gauge transformation is only valid in the limit of weak chiral symmetry breaking.

This distinction in the Majorana wavefunction in convex or non-convex shaped topological superconductors once again shows the influence which the shape of a topological superconductor has on the Majorana localization.

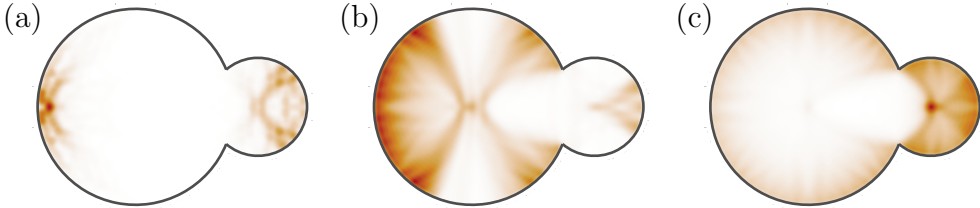

Figure 6: Local density of the Majorana wavefunction of a p-wave topological super-conductor billiard comprised of two circles of radii $100a$ and $50a$, with their centers separated by $80a$. The parameters of the system are: $\mu = 0.61t$, $\Delta_x = 0.07ta$ and $\Delta_y = 0ta, 0.015ta, 0.07ta$ for (a)/(b)/(c). The wavefunctions has been smoothed by convoluting it with a bell-shaped bump function.

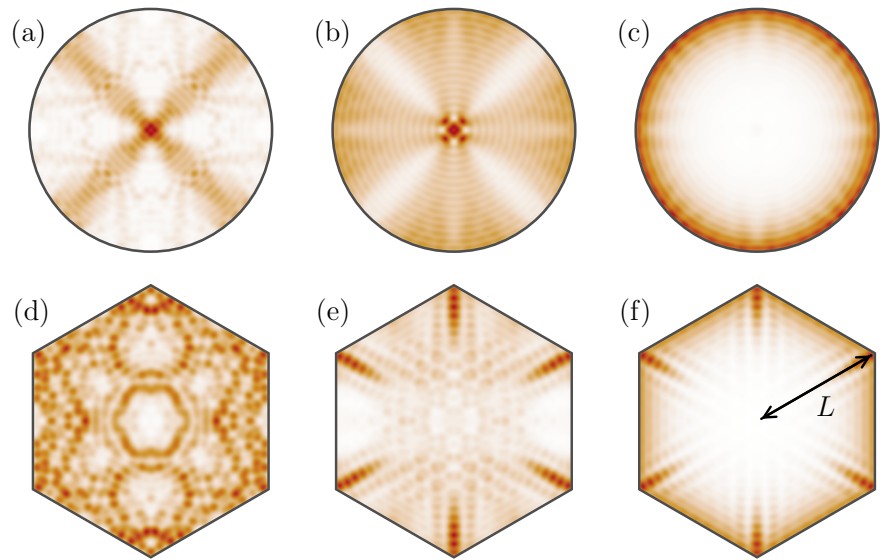

Figure 7: Caustic formation in the Majorana wavefunctions in the presence of chiral symmetry breaking. Panels (a), (b) and (c) show the Majorana wavefunction density in a billiard comprised of a circle of radius $80a$. (d), (e) and (f) show the Majorana wavefunction density for a hexagon with $L = 80a$. All wavefunctions are obtained for $\mu = 0.61t$ and $\Delta = \Delta_x = \Delta_y = 0.032ta, 0.08ta, 0.2ta$ for (a)/(d), (b)/(e) and (c)/(f). The wavefunction densities have been smoothed by convoluting it with a bell-shaped bump function.

## 6 Majorana billiards in s-wave topological superconductors

In the previous sections, we have primarily focused on the simplest example of a topological superconductor, which is the spinless p-wave superconductor. However, the majority of well-known superconductors are of the s-wave type, where both spin species are present. An s-wave superconductor in close proximity to a semiconductor with large Rashba spin-orbit interaction and a Zeeman field effectively gives rise to a p-wave topological superconductor [42–44]. For completeness, we investigate whether the shape of the system similarly affects the Majorana wavefunction of an s-wave topological superconductor.

We consider the Hamiltonian of a two dimensional electron gas with spin-orbit interaction and Zeeman field, proximitized by an s-wave superconductor:

$$H_s = h(\mathbf{p},\mathbf{r})\tau_z + (\alpha_x p_x \sigma_y - \alpha_y p_y \sigma_x)\tau_z + B\sigma_z + \Delta\tau_x\,, \tag{11}$$

where $\alpha_{x,y}$ is the spin-orbit coupling strength, $B$ is the Zeeman energy, $\tau_i(\sigma_i)$ denote the Pauli matrices in the particle-hole (spin) basis and $h(\mathbf{p},\mathbf{r}) = \mathbf{p}^2/(2m) - \mu + V(\mathbf{r})$ is the single-particle Hamiltonian with $\mu$ being the chemical potential and $V(\mathbf{r})$ being the confinement potential energy. For an s-wave topological superconductor, changing either the Zeeman energy $B$ or the chemical potential $\mu$ may lead to a switch in the ground state fermion parity. Without loss of generality, we choose to vary $\mu$ and keep $B$ constant while ensuring that the condition for the topological phase is satisfied, $B > \sqrt{\Delta^2 + \mu^2}$ [42]. The case $\alpha_x = \alpha_y = \alpha$ corresponds to pure Rashba spin-orbit splitting. The case $\alpha_x > 0, \alpha_y = 0$ can be reached either via the interference of Rashba and Dresselhaus spin-orbit couplings [45], or can be effectively realized if the confinement in $y$ direction is smaller than the spin-orbit length [46–48].

In the presence of chiral symmetry, we use the mapping described in Ref. [35] to find the zero energy solutions of the s-wave topological superconductor Hamiltonian given in Eq. (11). Specifically, we consider the case where $\alpha_x > 0$ and $\alpha_y = 0$. As we show in Appendix B, the Majorana wavefunction at a fermion parity switch (i.e. when there is an exact zero energy solution in the confined system) of an s-wave topological superconductor then has the following form:

$$\phi_\pm(\mathbf{r}) = \sum_n \zeta_\pm(\epsilon)e^{\pm x/\xi}\psi_n(\mathbf{r},\epsilon) + \zeta_\pm(-\epsilon)e^{\mp x/\xi}\psi_n(\mathbf{r},-\epsilon)\,, \tag{12}$$

where $\epsilon = \sqrt{B^2 - \Delta^2}$ and $\zeta_\pm(\epsilon)$ are the eigenvectors of the matrix $\epsilon\sigma_z \mp \Delta\sigma_x$. Here, $\psi_n(\mathbf{r},\pm\epsilon)$ are normal state wavefunctions of $h(\mathbf{p},\mathbf{r})\psi_n = \pm\epsilon\psi_n$ and $\xi = \hbar\epsilon/m\alpha_x\Delta$ is the coherence length. The term $e^{\pm x/\xi}$ causes the Majorana wavefunction to localize near the edges of the system. As $\alpha_x$ increases, the perturbative treatment we use in our mapping becomes less accurate. In this case, we find the corresponding normal state eigenfunctions by inspecting nearby states close to the target eigenfunction.

We now employ the following conceptual algorithm, similar to the one presented in Sec. 2 to show the correspondence between the s-wave Majorana wavefunction and the normal state eigenfunctions:

1. We numerically determine the chemical potential $\mu = \mu_c$, where the ground state fermion parity switches, as described in Appendix B.

2. We then find the Majorana wavefunctions of the s-wave topological superconductor.

3. As required by our mapping, we shift $\Delta_n = \Delta + m\alpha_x^2\Delta/\epsilon$ and $\mu_n = \mu + m\alpha_x^2\Delta^2/2\epsilon^2$, and solve for the normal state eigenfunctions around $\epsilon = \pm\sqrt{B^2 - \Delta^2}$. In the topological regime, when $\mu_c < \sqrt{B^2 - \Delta^2}$, the eigenstates with $\epsilon < 0$ are below the bottom of the normal state band. Therefore, we only consider eigenstates with $\epsilon > 0$.

4. Finally, we compare the Majorana wavefunction and the corresponding normal state eigenfunction.

Using this algorithm, we confirm that the Majorana wavefunction of a chaotically shaped s-wave topological superconductor also features quantum scarring. In Fig 8(a), we show a normal state eigenfunction in a stadium billiard which is more pronounced along the trajectories of a "candy" shaped quantum scar. Correspondingly, in Fig 8(b), we display the Majorana wavefunction, where an increased density is evident along the same trajectories as in the normal wavefunction. Increasing $\alpha_x$, the Majorana wavefunction becomes more localized towards

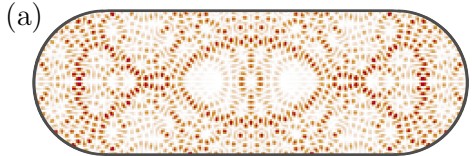
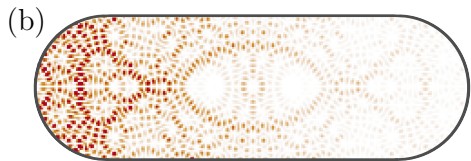

Figure 8: Correspondence between eigenfunctions of a normal state Hamiltonian and an s-wave superconductor Hamiltonian in identical stadium billiards with dimensions $L = 240a, W = 80a$. (a) shows a normal wavefunction density for $\mu = 0.57t$. (b) shows a s-wave wavefunction density for $\mu = 0.57t$, $\Delta = 0.25t, B = 0.8t$ and $\alpha_x = 0.045ta$. The Majorana wavefunction of s-wave topological superconductor is localized towards one direction as we put it in the basis of $\epsilon\sigma_z - \Delta\sigma_x$.

the edges as the coherence length decreases. This result confirms a similar outcome as in p-wave topological superconductors: the Majorana wavefunction reflects certain characteristics of the normal state eigenfunctions, thereby demonstrating the influence of the shape of the topological superconductor on the localization.[3]

Having established the connection for the chiral symmetric case, we now examine the case when the chiral symmetry is broken. Specifically, we consider an s-wave topological superconductor Hamiltonian with $\alpha_x = \alpha_y > 0$. In this limit, we do not have an explicit mapping to a normal state Hamiltonian as before. Therefore, we investigate the Majorana wavefunctions in differently shaped topological superconductors without any correspondence to normal state eigenfunctions. To that end, we first consider $\alpha$ values that would yield coherence length comparable to the system size. In Fig 9a) and b) we display a convex-shaped s-wave topological superconductor with a clear caustic formation in the Majorana wavefunction. In contrast, Fig 9c) depicts a non-convex-shaped s-wave topological superconductor, in which we do not observe caustics. Increasing $\alpha$ further such that the coherence length becomes much smaller than the system size, the Majorana wavefunction becomes a chiral edge state.

We remark that as we lack an analytical solution describing the Majorana wavefunction without chiral symmetry, we cannot predict the influence of the confinement potential $V(\mathbf{r})$ on the Majorana wavefunction. However, the presence or absence of caustics caused by the convexity of the billiard suggests that the conclusions we draw in the p-wave case carry over to the s-wave case: the shape of the topological superconductor has pronounced effects on the localization properties of the Majorana wavefunction.

## 7 Conclusion

In conclusion, we investigated the influence of the classical dynamics of the normal state Hamiltonian on the spatial positioning and localization of Majorana wavefunctions by studying topological superconductors of various shapes, among them archetypal billiards studied in the field of quantum chaos. We show how the Majorana wavefunction inherits the local properties of the underlying normal state eigenfunction, which in turn may exhibit properties from its underlying chaotic classical dynamics. We exemplified this phenomenon by studying quantum scar formation in the Majorana wavefunction in both chiral symmetric p-wave and s-wave topological superconductors. Moreover, we observe the formation of caustics in convex-shaped topological superconductors with broken chiral symmetry. Furthermore, we

---

[3]The Majorana wavefunction for the s-wave topological superconductors is in fact a superposition of two normal state eigenfunctions, see Eq. (12). We refer the reader to App. B for more detail.

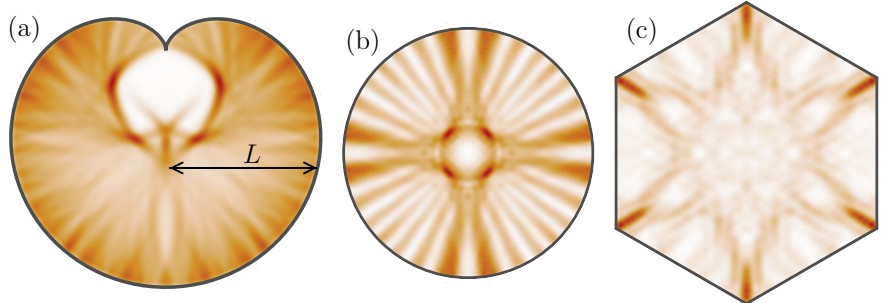

Figure 9: The s-wave wavefunction densities at $\mu = 0.29t$, $\Delta = 0.15t$, $B = 0.7t$, $\alpha_x = \alpha_y = 0.12ta$ in billiards of different shapes. Panel (a) shows a cardioid billiard with $L = 130a$. Panel (b) depicts a circle with radius $80a$. Billiard in (c) is a hexagon with $L = 80a$. The wavefunctions have been smoothed by convoluting it with a bell-shaped bump function.

demonstrated that a local potential far away from the edges of the system, hence far away from the localized Majoranas, can nevertheless change the local positioning and profile of their wavefunctions. We show how this phenomenon can be manipulated by altering the classical dynamics of the corresponding normal state.

The connection we made between classical dynamics and the Majorana wavefunction suggests a new source of localization that influences the Majorana wavefunctions. In addition to the pairing potential, ballistic chaotic localization also affects the local profile of the Majoranas, which can be of extreme importance in experimental setups in finitely sized topological superconductors. The effects of this new source of localization that we report in this paper are within experimental reach. It would be of interest to use these effects to understand and manipulate the localization properties of Majorana wavefunction. Finally, the mapping discussed here proves that many of the concepts developed in the field of semiclassical physics and quantum chaos will have their Majorana counterparts. Hence, it will be interesting to see whether or how these concepts play a role in the physics of Majorana zero modes.

## Acknowledgments

We acknowledge useful discussions with Anton R. Akhmerov, Klaus Richter, Steven Tomsovic, Juan Diego Urbina, Kim Pöyhönen, Antonio L. R. Manesco and Isidora Araya Day. İ. A. is a member of the Science Academy–Bilim Akademisi–Turkey; A. M. B. thanks the Science Academy–Bilim Akademisi–Turkey for the use of their facilities.

**Data availability** The code used to generate the figures is available on Zenodo [49].

**Author contributions** R. J. Z. and A. M. B. performed the numerical simulations. R. J. Z. prepared the figures. R. J. Z. and A. M. B. wrote the manuscript with input from M. W. and İ. A. A. M. B., M. W. and İ. A. defined the project scope. All authors analyzed the results. M. W. and İ. A. oversaw the project.

**Funding information** This work was supported by the Dutch Organization for Scientific Research (NWO) through OCENW.GROOT.2019.004.

# A   Chiral symmetry broken p-wave topological superconductor

We consider a 2D p-wave topological superconductor with broken chiral symmetry, described by the Hamiltonian:

$$H_p = h(\mathbf{p}, \mathbf{r})\,\tau_z + \Delta_x p_x \tau_x + \Delta_y p_y \tau_y\,, \tag{A.1}$$

where $\Delta_i$ is the anisotropic p-wave pairing potential strength, $h(\mathbf{p}, \mathbf{r}) = p^2/2m + V(\mathbf{r}) - \mu$ is the single-particle Hamiltonian with $\mu$ being the chemical potential and $V(\mathbf{r})$ is the potential which includes disorder and confinement potentials. $\tau_i$ are the Pauli matrices in particle-hole space ($i = x, y, z$).

Our aim is to find the set of chemical potential values $\mu_p$ for which the p-wave Hamiltonian has a zero-energy eigenstate:

$$H_p|_{\mu=\mu_p}\Psi = 0\,, \tag{A.2}$$

where $\Psi$ is the Majorana-spinor.

We map the problem of finding $\mu_p$ into a problem of finding real eigenvalues of a non-Hermitian operator. We premultiply the equation above by $\tau_z$ and obtain

$$\left(\frac{p^2}{2m} + V(\mathbf{r}) - \mu + i\Delta_x \tau_y p_x - i\Delta_y \tau_x p_y\right)\Psi = 0\,. \tag{A.3}$$

Taking the chemical potential to the right hand side of Eq. (A.3) and rearranging the terms, we obtain the following eigenvalue equation:

$$\left(\frac{\left(\mathbf{p} + i\boldsymbol{\eta}'\right)^2}{2m} + V(\mathbf{r}) + \frac{m}{2}\left(\Delta_x^2 + \Delta_y^2\right)\right)\Psi = \mu\,\Psi\,, \tag{A.4}$$

where $\boldsymbol{\eta}' = m\left(\Delta_x \tau_y \hat{x} - \Delta_y \tau_x \hat{y}\right)$. We then perform a similarity transformation to Eq. (A.4):

$$\tilde{\Psi} = U^{-1}\Psi\,, \tag{A.5}$$

$$\tilde{H} = U^{-1}HU\,, \tag{A.6}$$

where we choose $U = e^{\boldsymbol{\eta}'\cdot\mathbf{r}/\hbar}$. We compute the transformation of the Hamiltonian by examining how the individual terms in Eq. (A.4) transform:

$$U^{-1}V(\mathbf{r})U = V(\mathbf{r})\,, \tag{A.7}$$

$$U^{-1}\mathbf{p}U = \mathbf{p} - i\boldsymbol{\eta}'\,, \tag{A.8}$$

$$U^{-1}\boldsymbol{\eta}'U \approx \boldsymbol{\eta}' - i\frac{2m^2\Delta_x'\Delta_y'}{\hbar}(\hat{\mathbf{z}}\times\mathbf{r})\,\tau_z\,. \tag{A.9}$$

In the last line, we assume $L_i \ll \hbar/(m\Delta_i)$, where $L_i$ represents the system size for $i = x, y$. Using these result, we perform the similarity transformation and obtain the effective Hamiltonian:

$$\left(\frac{\left(\mathbf{p} + \frac{2m^2\Delta_x\Delta_y}{\hbar}(\hat{\mathbf{z}}\times\mathbf{r})\,\tau_z\right)^2}{2m} + V(\mathbf{r}) + \frac{m}{2}\left(\Delta_x^2 + \Delta_y^2\right)\right)\tilde{\Psi} = \mu\,\tilde{\Psi}\,. \tag{A.10}$$

We find that the zero-energy solutions for the p-wave Majorana billiard system are eigenvalues of the normal state Hamiltonian with a fictitious magnetic field $\pm 2m^2\Delta_x\Delta_y/e\hbar$ and a constant potential shift $\frac{m}{2}\left(\Delta_x^2 + \Delta_y^2\right)$.

We then calculate the cyclotron radius of this fictituous magnetic field:

$$r_c = \frac{mv}{eB} = \frac{\hbar m v}{2m^2 \Delta_x \Delta_y} \approx \frac{\pi \xi_x \xi_y}{\lambda_F}, \tag{A.11}$$

where $\lambda_F$ is the Fermi wavelength for a given chemical potential $\mu$. The bending of the trajectories due to the fictitious magnetic field becomes apparent in the semiclassical limit, i.e. $\lambda_F \ll L_i$. On the other hand, our mapping holds when $\xi_i \gg L_i$. As a result, the effective cyclotron radius of the fictitious magnetic field would always be larger than the system size, namely $r_c \ll L_i$.

# B    S-wave topological superconductor mapping

The superconducting Hamiltonian with s-wave pairing, Zeeman spin-splitting and Rashba spin-orbit coupling is as follows:

$$H_s = h(\mathbf{p}, \mathbf{r})\tau_z + (\alpha_x p_x \sigma_y - \alpha_y p_y \sigma_x)\tau_z + B\sigma_z + \Delta\tau_x, \tag{B.1}$$

where $\alpha_{x,y}$ are the magnitude of the spin-orbit coupling, $B$ the Zeeman energy, $\tau_i(\sigma_i)$ refer to the Pauli matrices in the particle-hole (spin) basis and $h(\mathbf{p}, \mathbf{r}) = \mathbf{p}^2/(2m) - \mu + V(\mathbf{r})$. Similarly to the p-wave case in Sec. 2, we look for Majorana zero modes which are described by the zero energy solutions:

$$H_s|_{\mu_i, B_i}\Psi = 0. \tag{B.2}$$

As both $\mu$ and $B$ can be tuned to satisfy this equation, we make the choice to fix $B$ and vary $\mu$. Then, to find the $\mu$ which corresponds to a fermion parity switch, we multiply Eq. (B.1) by $\tau_z$ from the left hand side and obtain:

$$\frac{\mathbf{p}^2}{2m} + V(\mathbf{r}) + (\alpha_x p_x \sigma_y - \alpha_y p_y \sigma_x) + B\sigma_z \tau_z + i\Delta\tau_y)\Psi = \mu\Psi, \tag{B.3}$$

where we have also taken $\mu$ to the right hand side. The fermion parity switches occur when the eigenvalues $\mu$ of this non-Hermitian eigenvalue equation are real. We numerically solve for the real eigenvalues of this non-Hermitian eigenvalue equation and obtain the chemical potential values where the fermion parity switches.

We once again consider the case where $H_s$ has chiral symmetry, choosing that $\alpha_x > 0$ and $\alpha_y = 0$. It is then possible to make Eq. (B.3) block off-diagonal in the particle-hole basis. After performing this, we obtain the following eigenvalue equation for the $2 \times 2$ blocks:

$$H_s \phi_\pm = [h(\mathbf{p}, \mathbf{r})\sigma_z - i\alpha p_x \sigma_x \mp (B + \Delta\sigma_x)]\phi_\pm = 0. \tag{B.4}$$

As a next step, we perform an imaginary gauge transformation on $\phi$ to perturbatively remove the imaginary Rashba-like term. Details of this transformation can be found in Ref. [11]. By using the fact that the resulting blocks of the Hamiltonian commute with each other to find the following eigenvector solutions:

$$\phi_\pm(\mathbf{r}) = \sum_n \zeta_\pm(\epsilon)e^{\pm x/\xi}\psi_n(\mathbf{r}, \epsilon) + \zeta_\pm(-\epsilon)e^{\mp x/\xi}\psi_n(\mathbf{r}, -\epsilon). \tag{B.5}$$

The solution has the following components: a spinor, an exponential tail and the wavefunctions $\psi_n$. These wavefunctions $\psi_n$ are the wavefunctions of the equation

$$h(\mathbf{p}, \mathbf{r})\psi_n = \epsilon_n \psi_n, \tag{B.6}$$

and are thus the same as the wavefunctions of a regular Hamiltonian in a potential $V(\mathbf{r})$.

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
