# Peer review of "Quantum Scars and Caustics in Majorana Billiards"

_SciPost Physics, doi:SciPost Phys. 17, 147 (2024)_

## Round 2 · Referee Report · Anonymous (Referee 1) · 2024-3-18

Strengths

The manuscript presents a novel localization phenomenon in Majorana billiards that is explained by connecting the scarred Majorana states to the eigenstates of the corresponding normal state system.

Weaknesses

The main weakness of the manuscript is that some parts of text would benefit more depth explanations.

Report

The authors study scarring in various Majorana billiards that they connect to the scarring in the normal state counterpart, even demonstrating a way to manipulate these states with a localized perturbation. However, some parts of the text would benefit of further clarification. In conclusion, I can recommend the publication of this work in SciPost if the manuscript was adequately revised according to at least some of the mentioned issues.

Requested changes

I recommend the following clarifications/additions to the manuscript:

(1) The authors mention the Heller type scarring, and also refer to the many-body-scars. However, in this context, the authors could consider the perturbation-induced scarring [Phys.
Rev. Lett. 123, 214101 (2019); J. Phys. Condens. Matter 31, 105301 (2019); Phys. Rev. B 96, 094204 (2017)] as well. As I see, there should be a very straightforward way to generalize the Majorana description to this type of scarring by considering locally perturbed p-type superconductors.

(2) The Hamiltonian for a topological superconductor is given, but it would be beneficial if the context behind this Hamiltonian would be described in more detailed, at least to give a citation.

(3) The authors refer to the potential of the system, but it is not defined clearly. What I have deduced is that they consider various billiard systems with hard wall boundaries where the potential is zero.

(4) It is unclear what the authors mean by the semiclassical limit. As far I see, all the simulations and analysis are fully quantum, and close to the ground state. Furthermore, the authors emphasize the topological nature of the superconductor, but not clarify if it has an impact the observed scarring, for instance whether the scars are chiral also (based on my conclusion, they are not).

(5) The authors consider how a hard disk stopper affects the scarred states, and mention this as a possible experimental avenue. However, a realistic STM nanotip would instead produce a soft bump. Nevertheless, I don't see this modification to change their conclusion.

(6) Finally, the manuscript mention multiple times the chaotic behavior of the system. However, no studies is carried out or shown, such as Poincare's surface of sections, or level statistics. In particular, I suspect the system to be highly mixed in the case when the artificial vector potential is present. For example, the candy wrap shape seen in Fig. 7 is not an periodic in a hard wall stadium, but it does appear in a smooth stadium that appear more like an elliptical oscillator.

---

## Round 2 · Referee Report · Anonymous (Referee 2) · 2024-4-15

Strengths

1- Detailed discussion of the mapping between the wavefunctions of topological superconductor and normal state eigenfunctions.

2- Study of the effect of the confinement one the properties of superconductor wavefunctions

3- The paper is clearly written

Weaknesses

1- The results for the localization of superconductor wavefunctions often lack quantitative analysis.

2- The fact that what is seen on Fig 5 and 6 are caustic associated with the "effective magnetic field " of Eq (9) is not demonstrated.

3- The importance of these results for the physics of topological superconductor might be discussed further.

Report

In this paper, the authors make use of the mapping between the wave function of topological superconductors at parity switch and normal state Hamiltonian to study the localization properties of Majorana wavefunctions. The question addressed is interesting and the paper clearly written.

The authors however often stop their analysis to this mapping. For instance in Fig 2, it is clear that (b) is deduced from (a) through the mapping (5) and one can recognize visually the trace of the periodic orbit shown on (a). But does this really imply localization ? One would like to see a quantitative comparison of, for instance, inverse participation ratio, to see how much of the localization implied by scarring transfer to the Majorana through (5).

In the same way, the fact that a stopper placed in a location where the Majorana wavefunction is almost zero affect this latter is very reminiscent of Aharonov-Bohm effect, and maybe more discussion about how much this analogy is valid could be useful.

Also, although it is true that Berry discuss in [32] the consequence of caustic for the "magnetic Green function", this is the free Green function which is discussed there (without confining potential). This of course would also apply qualitatively to a billiard if the cyclotron radius is much smaller than the typical size of the system. However the regime considered here (weak chiral symmetry breaking) seems to imply weak effective magnetic field, and I am afraid that what we see might just be the caustics of the unperturbed dynamics. This in any case has to be clarified.

Finally, as non-specialist of topological superconductor, I would have appreciated more in depth discussion about the importance, for topological superconductor physics, of the kind of localization discussed in the paper.

Requested changes

1-Study more quantitatively the localization properties of the scarred wavefunctions, by comparing for instance their inverse participation ration to the average one.

2- Provide information about the cyclotron radius for the effective magnetic field in Fig 5 and 6 and discuss how much this effective magnetic field modifies the classical dynamics. More generally show which classical trajectories (and thus which caustics) are involved in these figure.

3- Discuss the connection (or absence of connection) between the phenomenology observed in Fig. 3 & 4 and Aharonov-Bohm effect.

4- Provide more discussion about the importance, for topological superconductor physics, of the kind of localization discussed in the paper.

Point of details :

a- As the Majorana wavefunctions are 1/2 spinors, it might be useful to specify to what correspond the scalar functions plotted in the various figures.

b- Figure 2c is a rather trivial consequence of the the mapping (5). Is this really useful ? In any case putting it next to Fig 2a&b could give the wrong impression that it's a check of the localization of the wavefunction (due to scarring), and is thus slightly misleading.

c- Just before Eq (9) : why could we set \Delta_x = \Delta_y= \Delta "without loss of generality" ?

Recommendation

Ask for major revision

---

## Round 3 · Referee Report · Anonymous (Referee 1) · 2024-8-9

Report

The authors have replied to my questions and criticism with satisfactory, and I recommend the publication of the manuscript.

Recommendation

Publish (meets expectations and criteria for this Journal)

---

## Round 3 · Referee Report · Anonymous (Referee 2) · 2024-10-24

Report

The authors gave a satisfactory answer to most of the remarks I've made in my first report.

Recommendation

Publish (meets expectations and criteria for this Journal)

---

## Round 3 · Author Response

We thank both referees for their overall positive evaluation. We have implemented all the requested changes. We also provide the redlined manuscript with the changes highlighted at this URL: https://surfdrive.surf.nl/files/index.php/s/MS3BBCrYyU7Ae93.

Ref. 1:

(1) The authors mention the Heller type scarring, and also refer to the many-body-scars. However, in this context, the authors could consider the perturbation-induced scarring [Phys. Rev. Lett. 123, 214101 (2019); J. Phys. Condens. Matter 31, 105301 (2019); Phys. Rev. B 96, 094204 (2017)] as well. As I see, there should be a very straightforward way to generalize the Majorana description to this type of scarring by considering locally perturbed p-type superconductors.

We thank the referee for pointing out these references. Indeed, perturbation-induced scarring can also be relevant for p-wave topological superconductors. We have added these references in the introduction section of our paper.

(2) The Hamiltonian for a topological superconductor is given, but it would be beneficial if the context behind this Hamiltonian would be described in more detailed, at least to give a citation.

We thank the referee for their comment. Following their suggestion, we have added a reference to p-wave topological superconductor Hamiltonian we use in Eq. (1).

(3) The authors refer to the potential of the system, but it is not defined clearly. What I have deduced is that they consider various billiard systems with hard wall boundaries where the potential is zero.

We thank the referee for pointing out this possible source of confusion. We have now clarified what we meant by potential.

(4) It is unclear what the authors mean by the semiclassical limit. As far I see, all the simulations and analysis are fully quantum, and close to the ground state. Furthermore, the authors emphasize the topological nature of the superconductor, but not clarify if it has an impact the observed scarring, for instance whether the scars are chiral also (based on my conclusion, they are not).

Here, by semiclassical we mean that the system size is much larger than the wavelength of the electrons, which is set by the chemical potential values we use. We have now added a sentence that clarifies this point. About the second point raised by the referee, indeed these scars are not chiral, however, we believe this would be an interesting future direction of research.

(5) The authors consider how a hard disk stopper affects the scarred states, and mention this as a possible experimental avenue. However, a realistic STM nanotip would instead produce a soft bump. Nevertheless, I don't see this modification to change their conclusion.

The referee is right in pointing out that an STM nanotip would produce a softer potential. It is also true that this modification does not change the conclusion that the edge modes are affected by the potential produced by the potential far away from the edge.

(6) Finally, the manuscript mention multiple times the chaotic behavior of the system. However, no studies is carried out or shown, such as Poincare's surface of sections, or level statistics. In particular, I suspect the system to be highly mixed in the case when the artificial vector potential is present. For example, the candy wrap shape seen in Fig. 7 is not an periodic in a hard wall stadium, but it does appear in a smooth stadium that appear more like an elliptical oscillator.

We thank the referee for their comment. Here, we take advantage of the fact that the system shapes we focus on in this manuscript are known to feature chaotic dynamics. Figure 8 (Figure 7 in the previous version) indeed shows more complicated trajectories because the Majorana wavefunction is actually a superposition of two normal state wavefunctions, as shown in Eq. 12. To emphasize this point further, we have added the following footnote to our manuscript:

"The Majorana wavefunction for the s-wave topological superconductors is in fact a superposition of two normal state eigenfunctions, see Eq.12. We refer the reader to App. B for more detail."

Ref. 2:

The authors however often stop their analysis to this mapping. For instance in Fig 2, it is clear that (b) is deduced from (a) through the mapping (5) and one can recognize visually the trace of the periodic orbit shown on (a). But does this really imply localization ? One would like to see a quantitative comparison of, for instance, inverse participation ratio, to see how much of the localization implied by scarring transfer to the Majorana through (5). 1-Study more quantitatively the localization properties of the scarred wavefunctions, by comparing for instance their inverse participation ration to the average one.

We thank the referee for their comment. Following referee's recommendation, we have now included a new figure (Figure 3 in the new version) in our manuscript where we show that the inverse participation ratio for scarred and non-scarred wavefunctions at a finite $\Delta_x$ depend non-trivially on the initial state. We aim to quantatively show that the local structure of the underlying state carries over to localization of the majorana wavefunction.

..This of course would also apply qualitatively to a billiard if the cyclotron radius is much smaller than the typical size of the system. However the regime considered here (weak chiral symmetry breaking) seems to imply weak effective magnetic field, and I am afraid that what we see might just be the caustics of the unperturbed dynamics. This in any case has to be clarified. 2 - Provide information about the cyclotron radius for the effective magnetic field in Fig 5 and 6 and discuss how much this effective magnetic field modifies the classical dynamics. More generally show which classical trajectories (and thus which caustics) are involved in these figure.

We thank the referee for their comment. We have checked the cyclotron radius of the effective magnetic field and observed that there is no direct relation to the size of the causitcs we find. Although our theory is valid for small $\Delta_i$, our mapping deviates when $\Delta_i$ is larger, where the cyclotron radius becomes comparable to the system size. On the other hand, the appearance of caustics depends on having convex shapes and breaking chiral symmetry, which means both $\Delta_x$ and $\Delta_y$ must be non-zero. To illustrate this, we have replaced the previous Figure 5 with a new figure (now Figure 6), showing that caustics appear even at very low $\Delta_y$ for a specific non-zero $\Delta_x$. Based on these observations, we have revised the text to explain that caustics are caused by chiral symmetry breaking, not the effective magnetic field.

3- Discuss the connection (or absence of connection) between the phenomenology observed in Fig. 3 & 4 and Aharonov-Bohm effect.

We thank the referee for their comment. Despite being in a similar spirit with Aharonov-Bohm effect, Figures 4 and 5 (previously Figures 3 and 4) do not feature an orbital field (or effective orbital field that is present in the chiral symmetry broken case). Here, we consider topological superconductors with chiral symmetry. The phenomenology observed in these figures are due to mixing of the momentum states as a a result of the scatterer at the center of the billiard.

4- Provide more discussion about the importance, for topological superconductor physics, of the kind of localization discussed in the paper.

We thank the referee for their remark. In addition to the discussion we provide in our introduction and conclusion part, we now provide a discussion about importance of the ballistic localization on the localization properties of Majorana wavefunctions in the conclusion part of our manuscript. Specifically, we have added the following sentence in our manuscript:

"In addition to the pairing potential, ballistic chaotic localization also affects the local profile of the Majoranas, which can be of extreme importance in experimental setups in finitely sized topological superconductors."

a- As the Majorana wavefunctions are 1/2 spinors, it might be useful to specify to what correspond the scalar functions plotted in the various figures.

We thank the referee for pointing out this lack of information. We have now included a description of the density of the wavefunctions in caption of Figure 1: $\rho(\mathbf{r}) = \Psi^\dagger(\mathbf{r})\Psi(\mathbf{r})$.

b- Figure 2c is a rather trivial consequence of the the mapping (5). Is this really useful ? In any case putting it next to Fig 2a&b could give the wrong impression that it's a check of the localization of the wavefunction (due to scarring), and is thus slightly misleading.

We thank the referee for their remark. Our purpose of having Figure 2c is two-fold. Firstly, it demonstrates that our theory and numerical simulations match. Secondly, it serves as an example that shows the effect of the pairing potential $\Delta_x$ on the Majorana wavefunctions. To avoid any confusion, we have modified the text referring to Fig. 2c:

"Fig.2c) shows the localization of the Majorana wavefunction as a function of the pairing potential $\Delta_x$. To this end, we plot the logarithm of the normalized overlap between the Majorana wavefunction at $\Delta_x=0$ and the Majorana wavefunction at finite $\Delta_x$ for different $x$ values in the stadium billiard:"

c- Just before Eq (9) : why could we set \Delta_x = \Delta_y= \Delta "without loss of generality" ?

We thank the referee for their comment. Here, we mean that even if $\Delta_x\neq\Delta_y$, our mapping still holds. For completeness, we have now extended our analytical calculation to include $\Delta_x\neq\Delta_y$ and wrote a new appendix (Appendix A in the new version) that includes the details of the calculation.

---

## Round 3 · List of Changes

Difference file with list of changes included above.

---

## Editorial Decision

published